# Salinity, Temperature and Ammonia Acute Stress Response in Seabream (*Sparus aurata*) Juveniles: A Multidisciplinary Study

**DOI:** 10.3390/ani11010097

**Published:** 2021-01-06

**Authors:** Matteo Zarantoniello, Martina Bortoletti, Ike Olivotto, Stefano Ratti, Carlo Poltronieri, Elena Negrato, Stefano Caberlotto, Giuseppe Radaelli, Daniela Bertotto

**Affiliations:** 1Department of Life and Environmental Sciences, Polytechnic University of Marche, I-60131 Ancona, Italy; matteo.zarantoniello@gmail.com (M.Z.); i.olivotto@univpm.it (I.O.); rattistefano92@gmail.com (S.R.); 2Department of Comparative Biomedicine and Food Science (BCA), University of Padova, Viale dell’Università 16, I-35020 Legnaro, Padova, Italy; martina.bortoletti@phd.unipd.it (M.B.); carlo.poltronieri@unipd.it (C.P.); elena.negrato@unipd.it (E.N.); daniela.bertotto@unipd.it (D.B.); 3Valle Ca’ Zuliani Società Agricola Srl, I-34074 Monfalcone, Gorizia, Italy; stefano.caberlotto@vallecazuliani.it

**Keywords:** RT-PCR, immunohistochemistry, cortisol, IGFs, *mstn*, HSP70, *gr*, gilthead seabream, stress response, fish welfare

## Abstract

**Simple Summary:**

Gilthead seabream (*Sparus aurata*) is a teleost fish of great relevance in marine aquaculture, especially in the Mediterranean area and one of the most important farmed food marine species in Europe. Nevertheless, in captivity fish are usually exposed to stressful conditions, with a consequent negative impact on animal welfare and growth. The principal goal of this study was to shed light on the acute stress response of gilthead seabream juveniles exposed to different stressors using a multidisciplinary approach. For this purpose, gilthead seabream have been exposed to three different stress tests (temperature, salinity, and ammonia changes) and several laboratory techniques have been used to evaluate their growth and stress response. Results revealed that all the tested stressors had an impact on fish growth and health, particularly thermic and chemical exposure, whereas salinity seems to have a minor effect since this species can efficiently face with extreme variations in environmental salinity. The present work aimed to obtain relevant information on acute stress response of gilthead seabream to be used for improving farming condition and ensuring fish welfare.

**Abstract:**

The present study aimed to investigate the acute response of gilthead seabream (*Sparus aurata*) juveniles exposed to temperature, salinity and ammonia stress. Radioimmunoassay was used to evaluate cortisol levels, whereas insulin-like growth factors (*igf1* and *igf2*), myostatin (*mstn*), heat-shock protein 70 (*hsp70*) and glucocorticoid receptor (*gr*) gene expression was assessed trough Real-Time PCR. The presence and localization of IGF-I and HSP70 were investigated by immunohistochemistry. In all the stress conditions, a significant increase in cortisol levels was observed reaching higher values in the thermic and chemical stress groups. Regarding fish growth markers, *igf1* gene expression was significantly higher only in fish subjected to heat shock stress while, at 60 min, *igf2* gene expression was significantly lower in all the stressed groups. Temperature and ammonia changes resulted in a higher *mstn* gene expression. Molecular analyses on stress response evidenced a time dependent increase in *hsp70* gene expression, that was significantly higher at 60 min in fish exposed to heat shock and chemical stress. Furthermore, the same experimental groups were characterized by a significantly higher *gr* gene expression respect to the control one. Immunostaining for IGF-I and HSP70 antibodies was observed in skin, gills, liver, and digestive system of gilthead seabream juveniles.

## 1. Introduction

The worldwide decline of marine fisheries stocks has provided stimulus for rapid growth in fish and shellfish farming. Particularly, global production of gilthead seabream (*Sparus aurata*) has reached 237,049 tonnes in 2018 primarily from aquaculture activities [1]. In the Mediterranean area, the production of seabream ranges from extensive polyculture (e.g., vallicoltura in Italy and lagoon production in Egypt) or semi-intensive production in earth ponds (Portugal and southern Spain) to highly intensive land-based systems (raceways or tanks), inshore (Greece and Turkey) and offshore sea cages (Cyprus, Italy and Spain). 

An extensive literature is available about fish biology of stress and physiological and behavioural responses to a wide variety of physical, chemical and biological stressors, both in wild and captivity conditions (including aquaculture) [2,3,4,5,6]. In particular, water quality is one of the most important contributors to fish welfare; temperature, salinity and ammonia represent, in fact, the most common water parameters affecting physiological stress. Thermal and osmotic stress take place when water temperature or salinity, respectively, exceed the optimal ranges, modifying the normal physiological functions and triggering energy-consuming stress responses [7]. 

As a result of global climate change, it is predicted that variations in temperature and salinity could be important causes of stress in aquaculture [8,9], possibly affecting fish production. Moreover, temperature and salinity are considered abiotic crucial factors which may affect the development and survival of fish during larval growth [10,11,12]. 

In addition, rises in water temperature, salinity and pH led to ammonia concentration increase [7]. Ammonia is the main end product of nitrogen metabolism in marine teleosts and reduced growth was observed in fish exposed to increased ammonia levels for a short period [13,14].

Variations in water quality parameters (temperature, salinity, ammonia content) can induce fish growth reduction influencing the expression of insulin-like growth factors (*igf1* and *igf2*) and myostatin (*mstn*) [14,15,16,17,18,19] by the activation of stress response system. In fish, both IGF complex and myostatin play a key role in growth regulation [10,20,21,22,23]. The IGF complex includes the two highly conserved primary ligands (IGF-I and IGF-II), high-affinity transmembrane receptors that belong to the insulin/IGF receptor family and six IGF-binding proteins (IGFBP-1 to -6) [22,24,25]. In fish, IGF-I is mainly produced in liver, although numerous other organs express this molecule as well [26,27,28,29,30,31,32]. IGF-II shows a structural sequence similar to that of IGF-I and, in fish, exhibits a ubiquitous expression and acts mainly as a growth factor [29,33,34,35,36]. Myostatin is a member of the TGF-β superfamily and, in fish, its expression has been observed in several organs such as brain, eyes, exocrine and endocrine pancreas, gills, gonads, heart, intestine, kidney, liver, oesophagus, pharynx, skin, spleen, stomach and muscle fish explants [37,38,39,40,41,42]. 

The primary stress response in fish involves the release of catecholamines and activation of the hypothalamic–pituitary–interrenal (HPI) axis. Corticotropin releasing factor from the hypothalamus acts on the pituitary to synthesize and release corticotropic hormone, which in turn stimulates the synthesis and mobilization of glucocorticoid hormones (cortisol in teleost fish) from the interrenal cells [2,43]. Cortisol acts as an agonist of glucocorticoid receptor (GR). GR is a member of nuclear hormone receptor superfamily of ligand-activated transcription factors and it regulates gene transcription interacting with glucocorticoid response elements (GREs) or with numerous cytosolic proteins including chaperones, kinases, phosphatases and proteasome [44] to recover homeostasis and HPI axis functioning [45,46].

At cellular level, the stress response is mediated by the heat shock proteins (HSPs), a family of highly conserved proteins that are present in all cells in all life forms [3,47,48,49]. In fish, a variation in HSP70 expression has been observed not only after exposure to thermal shock [3], but also as a consequence of an osmotic shock [50,51,52] or after an ammonia stress exposure [14,53,54]. 

In this study, the effect of acute changes in temperature, salinity and ammonia, within a range that is likely to be found in a fish farm, were investigated in gilthead seabream juveniles. Particularly, the role of these abiotic factors on growth markers and short-term stress response was investigated through a multidisciplinary approach including: (i) Radioimmunoassay (RIA) to evaluate cortisol levels; (ii) Real-time PCR to evaluate *gr*, *hsp70*, *igf1*, *igf2* and *mstn* gene expression (iii) immunohistochemistry to evaluate the cellular localization of HSP70 and IGF-I. 

## 2. Materials and Methods

### 2.1. Ethics

All procedures involving animals were conducted in line with the Italian legislation and approved by the Ethics Committee of Università Politecnica delle Marche and the Italian Ministry of Health (prot. No AQUASTRESS/2013). 

### 2.2. Experimental Design and Sampling 

Gilthead seabream juveniles (*n* = 600; 70 dph; mean weight 0.25 g) were provided by the Valle Cà Zuliani Società Agricola s.r.l. (Monfalcone, Go) and transported to the Department of Life and Environmental Sciences (Ancona) in aerated tanks.

Fish have been acclimated for 25 days in four 200 L tanks equipped with mechanical biological and UV filtration (Panaque, Viterbo, Italy) and fed (2% body weight) with a commercial diet (Perla 4.0–5.0, Skretting) three times a day before the stress tests. Water parameters were kept constant and daily controlled (temperature 19 °C; salinity 30 ppt; ammonia: <0.02 mg/L; nitrates 10 mg/L). Finally, 24 h before the stress test, fish were starved and transferred in three 100 L tanks per experimental treatment (control, salinity, temperature and ammonia). 

Fish were then subjected to lower salinity (20‰), higher temperature (28 °C) and higher ammonia concentration (1.5 mg/L), respectively. The degree of changes of each stressor has been selected on the basis of adverse conditions expected in a commercial farm. Temperature increase (2 °C/10 min) was obtained by the use of a 1000 W heaters (Prodac, Cittadella, Padova, Italy), while salinity was decreased by gradually adding distilled water (2 ppt/min). High ammonia concentration was gained by adding water obtained by macerating commercial feed over the days before the stress test (20 L/10 min). The beginning of the test was set at the reach of the stress conditions for each experimental group (20‰ of salinity, 28 °C of water temperature and 1.5 mg/L of ammonia, respectively).

At 0–30 and 60 min from the beginning of the test, 10 fish per tank (30 fish per experimental group) were euthanized with a lethal dose of MS222 (300 mg/L) and properly stored for further analyses.

### 2.3. Cortisol

For cortisol analyses, 10 whole body fish per experimental group were immediately frozen in liquid nitrogen and stored at −80 °C. Whole body cortisol was measured by a specific microtiter radioimmunoassay (RIA) as described by Bertotto et al. [18]. Each fish was thawed out and pulverized in liquid nitrogen, and 100 mg of the resulting powders were suspended in 1 mL phosphate buffer (PBS, pH 7.2) and extracted with 8 mL of diethyl ether. The dry extracts were dissolved in PBS and used for RIAs. Briefly, a 96-well microtiter plate (Optiplate, Perkin Elmer Life Sciences) was coated with anti-rabbit c-globulin serum raised in a goat (dilution 1:1000 in 0.15 mM sodium acetate buffer, pH 9, at 4 °C) and, after PBS double washing, incubated overnight at 4 °C with the specific antiserum solution. Standards, quality controls, unknown extracts and 3H tracers were then added and, after overnight incubation at 4 °C, the plate was washed with PBS, added with 200 µL scintillation cocktail (Microscint 20, Perkin Elmer Life Sciences) and counted on a beta-counter (Top-Count, Perkin Elmer Life Sciences). 

The anti-cortisol serum showed the following cross-reactions: cortisol 100%, prednisolone 44.3%, 11-deoxycortisol 13.9%, cortisone 4.95%, corticosterone 3.5%, prednisone 2.7%, 17-hydroxyprogesterone 1.0%, 11-deoxycorticosterone 0.3%, dexamethasone 0.1%, progesterone < 0.01%, 17-hydroxypregnenolone < 0.01%, pregnenolone < 0.01%. 

To validate steroid determination in seabream juveniles’ whole body, competitive dose-response binding curves were created by serial extract dilutions of seabream (parallelism test) and the intra-inter-assays and recovery tests were performed. 

### 2.4. RNA Extraction and cDNA Synthesis

RNA extraction was performed according to Piccinetti et al. [55]. Total RNA extraction from 5 seabream juveniles’ whole body (randomly collected from each of the 3 tanks) was optimized using RNAzol RT reagent (Sigma-Aldrich, Saint Louis, MO, USA, R4533) following the manufacturer’s protocol. The total RNA extracted was eluted in 40 μL of RNase-free water (Qiagen). Final RNA concentrations were determined by NanoPhotometerTM P-Class (Implen, München, Germany). RNA integrity was verified by ethidium bromide staining of 28S and 18S ribosomal RNA bands on 1% agarose gel. RNA was stored at −80 °C until use. Total RNA was treated with DNAse (10 IU at 37 °C for 10 min, MBI Fermentas). Finally, 5 μg of RNA were used for cDNA synthesis using the iScript cDNA Synthesis Kit (Bio-Rad, Milan, Italy).

### 2.5. Real-Time PCR

PCRs were performed with SYBR Green in an iQ5 iCycler thermal cycler (both from Bio-Rad, Hercules, CA, USA), in triplicate according to Maradonna et al. [56]. Reactions were set on a 96-well plate by mixing, for each sample, 1 μL cDNA diluted 1:20, 5 μL of 2× concentrated iQ TM SYBR Green Supermix containing SYBR Green as the fluorescent intercalating agent, 0.3 μM forward primer and 0.3 μM reverse primer. The thermal profile for all reactions was 3 min at 95 °C, followed by 45 cycles of 20 s at 95 °C, 20 s at 60 °C and 20 s at 72 °C. Fluorescence was monitored at the end of each cycle. Dissociation curve analysis showed a single peak in all cases. 

Relative quantification of the expression of genes involved in fish stress response (*gr* and *hsp70*) and growth (*igf1*, *igf2* and *mstn*), was performed using β-actin and 18s as the housekeeping genes to standardize results by removing variation in mRNA and cDNA quantity and quality [57]. Amplification products were sequenced, and homology was verified. No amplification product was detected in negative controls and no primer-dimer formation was found in control templates. Data were analysed using the iQ5 optical system software version 2.0 including Genex Macro iQ5 Conversion and Genex Macro iQ5 files (all from Bio-Rad, Hercules, CA, USA). Primer sequences were designed using Primer3 (210 v. 0.4.0, Whitehead Institute for Biomedical Research, Cambridge, Massachusetts, USA) starting from seabream sequences available in GenBank. Primers were used at a final concentration of 10 pmol μL^−1^ (for sequences, please see Table 1).

### 2.6. Immunohistochemistry (IHC)

For immunohistochemistry, 5 seabream juveniles’ whole body per experimental group were fixed in 4% paraformaldehyde prepared in phosphate-buffered saline (PBS, 0.1 M, pH 7.4) at 4 °C overnight, washed in PBS, dehydrated through a graded series of ethanol and embedded in paraffin. Consecutive sections were cut at a thickness of 4 μm using a microtome (Leica, Wetzlar, Germany). Immunohistochemical staining was performed using the Elite ABC KIT system (Vector Laboratories, Inc., Burlingame, CA, USA). Before applying the primary antibodies, endogenous peroxidase activity was blocked by incubating the sections in 3% H_2_O_2_ in PBS. Non-specific binding sites were blocked by incubating the sections in normal goat serum (Dakocytomation, Milano, Italy). Sections were then incubated with the primary antisera (monoclonal anti HSP70 dilution 1:600, Stressgen Biotechnologies, San Diego, CA, USA; polyclonal anti IGF-1 dilution 1:200, Ibt System, Reutlingen, Germany), overnight at 4 °C. After washing with PBS, sections were incubated with biotin-conjugated anti-mouse or anti-rabbit Ig antibodies (Dakocytomation), washed with PBS and reacted with peroxidase-labeled avidin-biotin complex (Vector Laboratories). The immunoreactive sites were visualized using a freshly prepared solution of 10 mg of 3.3′-diaminobenzidine tetrahydrochloride (DAB, Sigma, Milano, Italy) in 15 mL of a 0.5 M Tris buffer at pH 7.6, containing 1.5mL of 0.03% H_2_O_2_. To ascertain structural details, sections were counterstained with Mayer’s haematoxylin.

The specificity of the immunostaining was verified by incubating sections with: (i) PBS instead of the specific primary antibodies; (ii) pre-immune sera instead of the primary antisera; (iii) PBS instead of the secondary antibodies. The results of these controls were negative (i.e., staining was abolished).

### 2.7. Statistical Analysis

Data from cortisol analysis have been subjected to analysis of variance by using a Linear Model of R software (R Core Team2019) with stress type and sampling time as main factors. Data from Real-time PCR analyses were analyzed by two-way ANOVA (with stress factor and sampling time as explanatory variable) followed by Tukey’s and Dunnett’s test for time course and sampling point analysis, respectively, with a statistical software package, SigmaStat 3.1 (Systat Software, Chicago, IL, USA). All data were expressed as the mean ± SD. Differences among means with *p* < 0.05 were accepted as being statistically significant.

## 3. Results

### 3.1. Cortisol

The cortisol assay showed acceptable parallelism and reproducibility (linear regression curve y = 19.2x − 0.2; regression coefficient R^2^: 0.99; CV % intra-assay = 5.8; CV % inter-assay = 13.9). The recovery test with value higher than 75% confirmed the efficiency of steroid extraction method. As reported in Figure 1, cortisol levels showed a significant increase related to the time in thermal and ammonia stress groups with respect to control both at 30 and 60 min (all *p* < 0.001) but not in the salinity group. In post stress sampling times, the salinity group cortisol level significantly differed from the others (*p* < 0.05 and *p* < 0.001) except to that of control at 60 min post stress. Significant differences were detected also between thermal and ammonia stress groups at 60 min (*p* < 0.001) but not at 30 min (Figure 1). 

### 3.2. Real-Time PCR Results

#### 3.2.1. Stress Response

The glucocorticoid receptor (*gr*) gene expression (Figure 2A) showed a time-related significant (*p* < 0.05) decrease in control group, from 0 to 60 min treatment. Salinity and thermic stress groups did not show significant changes in *gr* transcript during the trial, while ammonia stress groups showed a significant (*p* < 0.05) increase in *gr* gene expression after 60 min treatment. As regards differences within the same sampling point, at both 0 and 30 min, no significant differences were observed among the experimental groups. Differently, at 60 min, both thermic and ammonia stress groups showed a significant (*p* < 0.05) higher *gr* gene expression with respect to control and salinity stress groups (Figure 2A).

Considering *hsp70* gene expression (Figure 2B), control and salinity stress groups did not show significant changes neither in time nor within the same sampling points. During the time course, a statistically significant (*p* < 0.05) increasing of *hsp70* gene expression in both thermic (after 30 and 60 min) and ammonia (after 60 min) stress groups was observed. Considering differences within the same sampling time, a significantly (*p* < 0.05) higher *hsp70* gene expression was evident in the thermic stress (at both 30 and 60 min) and ammonia (only at 60 min) groups respect to the other ones. 

#### 3.2.2. Fish Growth Markers

As regards both *igf1* and *igf2* gene expression (Figure 3A,B), significant differences during the time course and within the same sampling time were detected only after 60 min treatment. In fact, both time course and time point analysis, evidenced a significant increase (*p* < 0.05) of *igf1* transcript in thermic stress group after 60 min treatment respect to the other sampling points (0 and 30 min) (Figure 3A). Differently, a significant increasing (*p* < 0.05) of *igf2* gene expression (Figure 3B) was observed in control and ammonia groups at 60 compared to 0 and 30 min of stress treatment. Furthermore, at 60 min, all the groups subjected to a stress condition showed a significantly (*p* < 0.05) lower *igf2* gene expression respect to control group. 

Regarding *mstn* gene expression (Figure 3C), time course analysis showed a significant (*p* < 0.05) increase after 30-min treatment followed by a significant decrease after 60-min treatment in thermic stress group, while in ammonia stress group, a significant (*p* < 0.05) increase of *mstn* gene expression was detectable only after 60-min treatment. Considering differences within the same sampling time, a significantly (*p* < 0.05) higher *mstn* gene expression was evident in the thermic stress (at both 30 and 60 min) and ammonia (only at 60 min) groups respect to the other ones.

### 3.3. Immunohistochemistry

#### 3.3.1. HSP70 Immunohistochemistry

In general, HSP70 immunoreactivity was expressed in skin, gills, liver and digestive system of both controls and stressed animals with a higher intensity in stressed animals than in controls. Instead, the lateral muscle, cartilages and connective tissue of control groups did not show any HSP70 expression (Table 2).

In particular, at cellular level, immunoreactivity to HSP70 antibody was detected in: (i) the epithelial cells of skin (Figure 4A); (ii) scattered cells of gill epithelium at the level of primary and secondary lamellae (Figure 4B); (iii) scattered cells of the gastric epithelium of stomach as well as in the gastric pits (Figure 4C); (iv) the rodlet cells scattered throughout the intestinal epithelium of ammonia stressed animals (Figure 4D). The hepatocytes of liver parenchyma showed an immunopositivity too (insert in Figure 4D). 

#### 3.3.2. IGF-I Immunohistochemistry

In general, an intense immunoreactivity was present in the skin, gills, and digestive system of stressed fish, whereas no reactivity was found in control animals at the level of trunk musculature, cartilages, and dense connective tissue, regardless of the stress experimental condition (Table 2). In particular, at cellular level, the anti-IGF-1 antibody revealed an immunostaining in: (i) the epithelial cells of skin (Figure 5A); (ii) scattered cells of gill epithelium at the level of primary and secondary lamellae (Figure 5B); (iii) the cytoplasm of cells lining the lumen of the stomach as well as in the cytoplasm of gastric pits (Figure 5C); (iv) the cytoplasm of enterocytes (Figure 5D). Moreover, rodlet cells imunoreactivity to IGF-I antibody was detected in the intestine epithelium of fish subjected to ammonia stress (Figure 5D). The parenchyma of liver showed an immunopositivity at the level of the cytoplasm of the hepatocytes (Figure 5D). 

## 4. Discussion

In aquaculture practices, temperature and salinity changes and increasing ammonia concentrations are considered important fish stressors [58]. Changes in temperature and salinity can affect metabolism, growth and survival of cultured fish while short-term ammonia exposure can cause oxidative stress and apoptosis and altered osmoregulatory function [59,60]. Regardless of the stress factor, the primary response is the activation of HPI axis which releases corticosteroid hormones like cortisol [2,61]. Cortisol is one of the most commonly used biomarkers to detect a stressful situation in fish [18,62,63,64], but it is well known that also gene expression of stress markers like *hsp70* and *gr* can be considered a prized indicator to better understand fish cellular and physiological status [65,66]. The determining role of cortisol in the stress response has been confirmed also in the present study. In fact, fish exposed to short-term changes of temperature and ammonia, similar to those that can possibly occur in a commercial farm, were characterized by higher levels of this hormone respect to control already after 30 min of treatment. Increases of cortisol due to changes in these two environmental factors have been already reported in other species as black seabream (*Spondyliosoma cantharus*), blunt snout bream (*Megalobrama amblycephala*), brook trout (*Salvelinus fontinalis*), turbot (*Scophthalmus maximus*) and common carp (*Cyprinus carpio*) [59,67,68,69,70]. Acute change in salinity failed to increase cortisol significantly confirming the euryhalinity of the gilthead seabream [71,72]. 

The cortisol results in the present study are supported by molecular ones that evidenced a significantly increased *gr* gene expression in the same groups (temperature and ammonia) at 60 min of treatment, further highlighting that cortisol acts through glucocorticoid receptors and elevates GR transcript levels [73,74]. Furthermore, to confirm the stress response, temperature and ammonia groups were characterized also by a significant time dependent increase in the *hsp70* gene expression at 60 min. In teleost fish, in addition to the HPI axis activation, acute stressors may act at a cellular level causing a higher synthesis of heat shock proteins in order to offer the cell a protection in stress conditions [75,76,77]. Among all the HSPs, the HSP70 inducible isoform is frequently activated by thermal stress and temperature-induced increases in *hsp70* gene expression have been detected in different fish species [18,78]. In the present study, increased temperature elevated the *hsp70* gene expression level already at 30 min confirming the sensitivity of this indicator to the thermal shock [3,18]. In accordance with our results, different studies reported increased HSP70 transcriptions in fish exposed to ammonia [59,79]. As regards salinity stress, no significant changes in *gr* and *hsp70* gene expression were evident with respect to control according to the observed cortisol levels. As previously suggested, this result is justified by the fact that *Sparus aurata* is an euryhaline teleost able to easily cope with extreme changes in environmental salinity [71,72]. 

The primary function of fish responses to variable stress factors is to adapt metabolism to face the energy requirement and to maintain homeostasis [80]. At this regard, it is well known that a physiological link between stress and growth-related genes exists [32]. Teleost species subjected to various stress factors usually evidence increased serum cortisol levels which have been associated to a suppression of the somatotropic axis [81,82,83]. The mechanisms through which glucocorticoids inhibit growth may involve the GH/IGF-I/IGFBP network [84]. For this reason, the influence of acute stress on *igf*s expression has been investigated in different fish species evidencing, in most cases, a downregulation in fish exposed to different stressors as, among them, temperature or salinity changes or high ammonia levels [17,82,85,86,87,88]. Accordingly, in the present study, *igf2* gene expression was significantly lower in groups subjected to a stress condition with respect to control after 60 min of treatment. Differently, *igf1* gene expression was significantly higher even if only in the thermic stress group. At this regard, it should be pointed out that *igf1* expression is regulated by GH hormone and increases with the onset of the post-natal growth [89,90]. Juveniles fish show faster growth rate with increased temperature which can be considered as the major external factor influencing growth [91]. On this regard, it has been demonstrated that, in fish under optimal nutritional status, temperature can increase plasma GH levels with a consequent stimulation of *igf1* gene expression [92,93]. Finally, the higher *mstn* gene expression observed in thermic and ammonia stress groups could be related to the higher cortisol level detected in the same experimental groups. Glucocorticoids in fact strongly regulate myostatin transcript levels in mammals via glucocorticoid response elements (GR Es) in the myostatin promoter, and this has been demonstrated in several fish species, suggesting a possible direct regulation of cortisol on muscle growth also in these animals [94,95,96,97].

Given the importance of HSP70 and IGF-I in describing fish growth and welfare [30,77,98], in the present work the localization of these proteins has been investigated by immunohistochemistry. Immunopositivity for HSP70 and IGF-I antibodies was detected in several tissues and organs of gilthead seabream juveniles, regardless of the experimental condition. Considering IGF-I, an intense reactivity was present in the skin, gills, and digestive system of stressed fish, whereas no reactivity was found in control animals at the level of trunk musculature, cartilages, and dense connective tissue. Immunopositivity was detected also in the parenchyma of liver. Specifically, a strong IGF-I immunoreactivity was evident in the epithelial cells of skin, stomach (including the gastric pits) and intestine, in the epithelium of both primary and secondary lamellae of gills, and in the hepatocytes of liver. These results are in agreement with those in literature [29,30,99,100,101], where a similar pattern of IGF-I immunostaining was observed in *Sparus aurata* and other fish species, thus attesting the role of IGF-I in the regulation of fish somatic growth, regardless of whether the animal is stressed or not. As to HSP70, it was expressed in skin, gills, liver and digestive system of both controls and stressed animals. Instead, the lateral muscle, cartilages and connective tissue of control groups did not show any HSP70 expression. The same immunohistochemical localization of HSP70 has been found by Fiocchi et al. [102] in European sea bass (*Dicentrarchus labrax*) exposed to manipulation and temperature changes. In fact, they observed both a weak presence of immunoprecipitates in the control groups and an increased immunopositivity in gills, liver and intestine of heat shock stressed fish. This notable increase of immunoprecipitates in the skin, gills, liver and intestine was evident also in other studies [77,103,104] of fish exposed to abiotic stress (environmental pollution and transport). Moreover, rodlet cells imunoreactivity to both IGF-I and HSP70 antibodies was detected in the intestine epithelium of fish subjected to ammonia stress. This result is consistent with those of Fiocchi et al. [102] from sea bass exposed to a different stressor, i.e., temperature, confirming that the appearance of these cells is commonly related to stressful conditions [102,103,105,106,107,108]. 

## 5. Conclusions

Short term exposure of seabream juveniles to changes in temperature and ammonia induced a stress response attested by a significant increase in cortisol, *gr* and *hsp70* gene expression levels within 60 min of treatment. The reduction in salinity did not evoke a significant response indicating the euryhaline characteristics of this species. Stress also produced variations in the transcripts of genes linked to growth but with less marked and clear trends, as is often reported in the literature. Considering the potential variability of the environmental factors tested in the farm, it will be necessary to carry out similar studies in the future to explore the response of animals after exposure to medium and long term stress conditions. 

## Figures and Tables

**Figure 1 animals-11-00097-f001:**
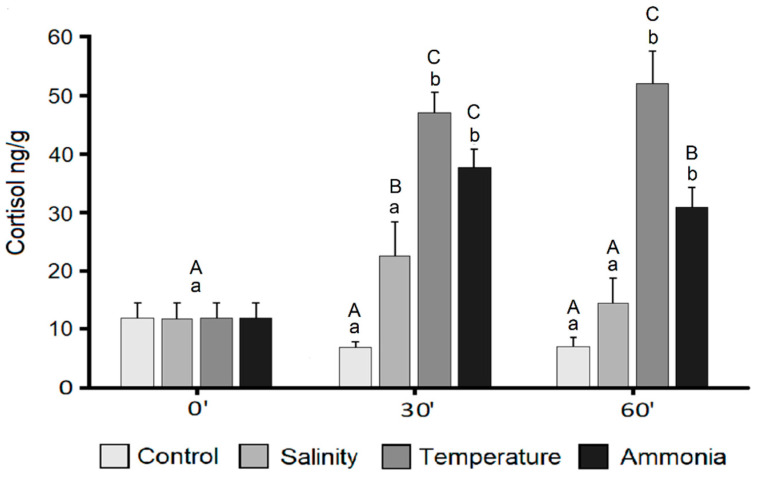
Cortisol concentrations in the different experimental groups of seabream juveniles at each sampling times (0, 30- and 60-min post stress). Data are expressed as mean ± SD (*n* = 10). Different letters denote statistically significant differences in time course analysis (lowercase) or within the same sampling point (uppercase) (*p* < 0.05).

**Figure 2 animals-11-00097-f002:**
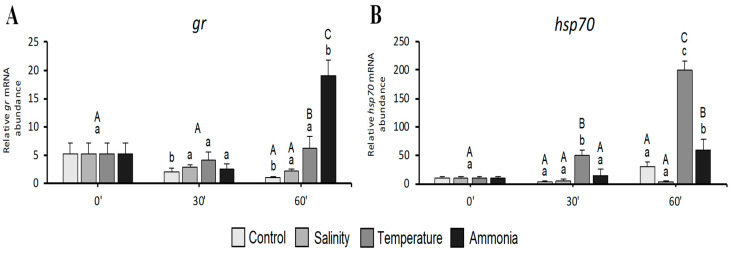
Relative mRNA levels of genes involved in stress response analysed in seabream juveniles: (**A**) *gr* and (**B**) *hsp70*. Data are expressed as mean ± SD (*n* = 15). Different letters denote statistically significant differences in time course analysis (lowercase) or within the same sampling point (uppercase) (*p* < 0.05).

**Figure 3 animals-11-00097-f003:**
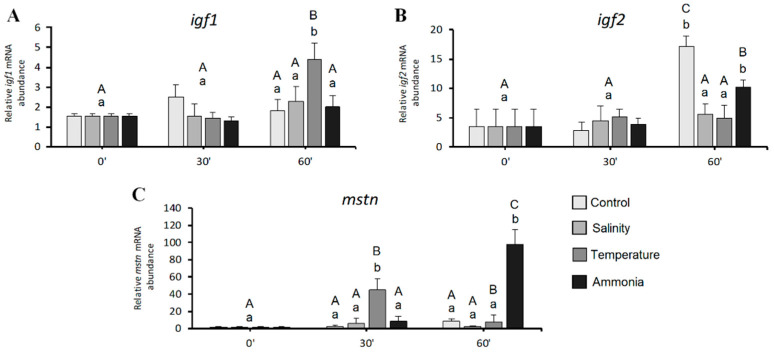
Relative mRNA levels of genes involved in fish growth analysed in seabream juveniles: (**A**) *igf1*; (**B**) *igf2* and (**C**) *mstn*. Data are expressed as mean ± SD (*n* = 15). Different letters denote statistically significant differences in time course analysis (lowercase) or within the same sampling point (uppercase) (*p* < 0.05).

**Figure 4 animals-11-00097-f004:**
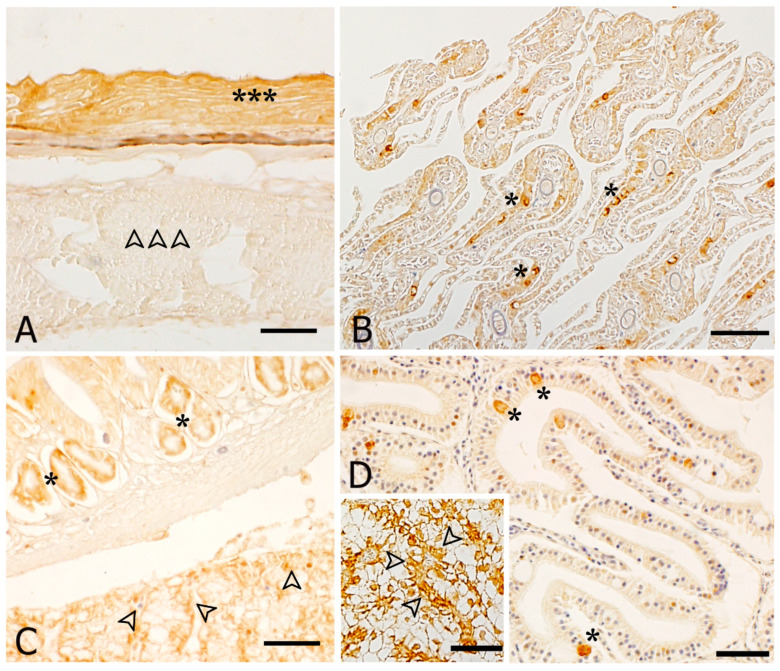
Immunohistochemical localization of HSP70 in *Sparus aurata*. All panels are counterstained with haematoxylin. (**A**) *control fish*. HSP70 immunoreactivity in the epithelium of the skin (asterisks); the lateral muscle is negative (arrowheads); (**B**) *control fish*. The epithelium lining both primary and secondary lamellae shows an immunoreactivity (asterisks); (**C**) *thermal stressed fish.* Gastric pits (asterisks) and liver parenchyma (arrowheads) exhibit an immunopositivity to HSP70 antibody; (**D**) *ammonia stressed fish* and as inserted a *salinity stressed fish.* Intestinal rodlet cells (asterisks) and hepatocytes are immunopositive (arrowheads). Bars: 200 μm for all panels.

**Figure 5 animals-11-00097-f005:**
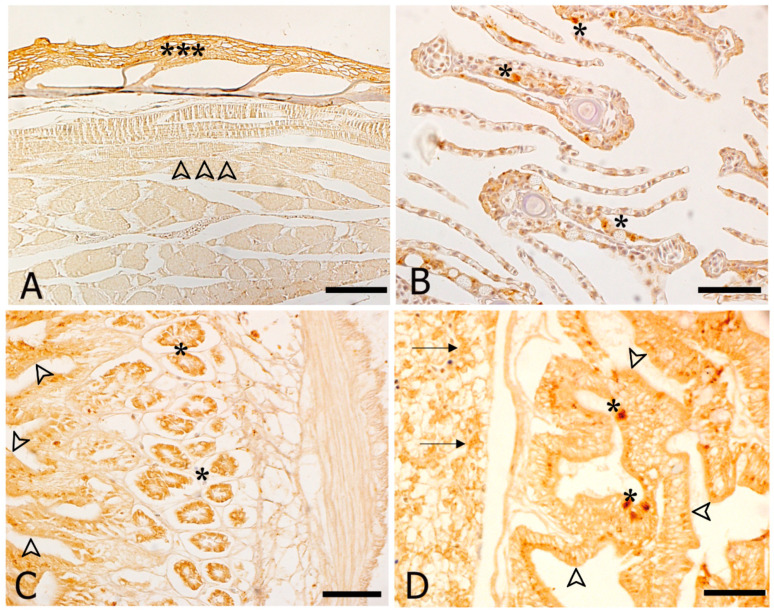
Immunohistochemical localization of IGF-I in *Sparus aurata*. All panels are counterstained with haematoxylin. (**A**) *control fish.* In the skin, immunopositivity was localized at the level of the epithelium (asterisks), whereas the trunk musculature is negative (arrowheads); (**B**) *thermal stressed fish.* In gills, the epithelium of primary and secondary lamellae showed an immunoreactivity (asterisks); (**C**) *salinity stressed fish*. The epithelium of the stomach shows immunoreactivity to IGF-I antibody (arrowheads); a positivity was also evident in the gastric pits (asterisks); (**D**) *ammonia stressed fish.* Enterocytes (arrowheads), intestine rodlet cells (asterisks) and liver parenchyma (arrows) showing strong immunoreactivity to IGF-I antibody. Bars: 40 μm (**A**), 200 μm (B), 200 μm (**C**), 200 μm (**D**).

**Table 1 animals-11-00097-t001:** Primers sequences used in this study.

Gene	Forward Primer (5′–3′)	Reverse Primer (5′–3′)
*gr*	5′- GCCTTTTGGCATGTACTCAAACC -3′	5′- GGACGACTCTCCATACCTGTTC -3′
*hsp70*	5′- GTACGGTCTGGACAAAGGCA -3′	5′- GGTTCTCTTGGCCCTCTCAC -3′
*igf1*	5′- AGCCCAGAGACCCTGTGC -3′	5′- CAGCTCACAGCTTTGGAAGCA -3′
*igf2*	5′- TGGGATCGTAGAGGAGTGTTGT -3′	5′- CTGTAGAGAGGTGGCCGACA -3′
*mstn*	5′- GGCCTGGACTGTGATGAGAA -3′	5′- GCATGTTGATGGGTGACATC -3′
*β-act*	5′- GGTACCCATCTCCTGCTCCAA -3′	5′- GAGCGTGGCTACTCCTTCACC -3′
*18s*	5′- GTGAGGTTTCCCGTGTTGAG -3′	5′- GACCATAAACGGTGCCAACT -3′

**Table 2 animals-11-00097-t002:** Immunohistochemical localizations of HSP70 and IGF-I in different tissues of *Sparus aurata* (control and stressed animals): −, not detectable; +/−, slight but above background levels; +, moderate staining; ++ marked staining.

Tissue	Ctrl		Salinity Stress		Temperature Stress		Ammonia Stress	
	HSP70	IGF-I	HSP70	IGF-I	HSP70	IGF-I	HSP70	IGF-I
Skin	+	++	++	++	++	++	++	++
Muscle	−	−	−	−	−	−	−	−
Gills	+	++	++	++	++	++	++	++
Stomach	+/−	++	+	++	++	++	+	++
Intestine	+/−	++	+	++	+	++	++	++
Liver	+/−	++	++	++	++	++	+	++

## Data Availability

The data presented in this study are available within the article. If needed, supplementary material is available on request from the corresponding author.

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
