# Peer review of "Salinity, Temperature and Ammonia Acute Stress Response in Seabream (Sparus aurata) Juveniles: A Multidisciplinary Study"

_animals, 2021, doi:10.3390/ani11010097_

Round 1

Reviewer 1 Report

The manuscript describes the effects of some physical stresses (salinity, ammonia, and temperature) on the HSP, cortisol, and IGF-system in seabream.  The authors performed qPCR to determined the gene expression of gr, IGF-1, IGF-2, etc.  Immunohistochemistry was also performed to determine the localization of HSP70 and IGF-1.  Radioimmunoassay was used to measure the body cortisol.  The experimental design was not satisfactory, and some improvements are suggested in the following.

Major comments

  1. The salinity treatment used in the present study may have little impact on the general physiology of seabream. Seabream is a physiological euryhaline species and it can survive a transfer from 30 ppt to 5-6 ppt.  The salinity treatment used in this study was 20 ppt, which is still hyperosmotic to the body fluid of seabream.  In fact, lower salinities at 12-15 ppt are considered as iso-osmotic, where the fish spend less energy on osmoregulation.  Therefore, 20 ppt is not exactly stressful physiologically.  The authors can try a gradual transfer of salinity down to 5-6 ppt (lower than the body fluid osmolality), so that the osmoregulatory epithelia must switch from ion-excretion to ion-absorption.
  2. Measurement of IGF-1 and IGF-2 has little physiological meaning as their levels are often not related to the local functions. In addition, the authors should explore the IGF-binding proteins, which modulate the functions (especially local functions) of IGF-1 and IGF-2.  Section 3.2.2 was titled as “Fish growth”.  However, the experiment finished in 60 min, and I do not see any reasonable claims of fish growth under this interval.  I suggest the authors to try a longer acclimation period to induce an observable effect on growth.
  3. The immunohistochemistry in Figure 4 and 5 was chaotic. The legends did not match with the figures.  I suppose A is skin, B is gill, C is stomach, and D is intestine.  For the effects of ammonia and thermal stresses, there are no comparisons with the intact control, nor is there any comparison at all among A-D.  The scale bar should be not correct, as 20 micron is the size of 2 red blood cells, and I think the magnification is way lower to give that resolution. 

Reviewer 2 Report

I really liked this manuscript. Your writing has clarity and conciseness, forming an article easy to comprehend. Results are discussed through an extensive review of references (sometimes in excess!).

However, results are on starting point to describe which are the most crucial stressors exerting their influence on captive fish and how stressors’ intensity and duration can modify this response. Similarly, the age of the stressed fish, the high variation of potential stressors in captivity, their cumulative influence but also the mechanisms lying beneath are of crucial importance. I hope that authors’ team will continue their effort to contribute valuable data, todays absent, for many fish species.

I have only some cautions on histological localization of hsp70 and igf1. In fact to be sure that stressor’ changes induce immunoporosivity of antibody, at a manner different than control fish, I think that we should compare sections of the same organ (also, from the same part of the organ) from control and stressed fish. If I have only a section form any organ of the stress fish, and the equivalent section from the stressed fish is missing I can’t reliably conclude.

I found that the article advances out knowledge on stress response of fish, it needs no revision,  thus I recommended to be accepted in its present form. 

Just to help, please find attached (animals-1026003-list-comments-authors)a short list of comments and modifications proposed for reviewing.

Reviewer 3 Report

The manuscript entitled” Salinity, temperature and ammonia acute stress response in seabream (Sparus aurata) juveniles: a multidisciplinary study” by Zarantoniello et al., is an elegant work regarding mainly the acute response of Gilthead seabream to different breeding conditions. The article is well written and the material and method are sound. Nevertheless, before publication, there is a minor issue that should be clarified. The Authors might try to identify the cellular type displaying the immunolabeling for the antibodies used.

Round 2

Reviewer 1 Report

Point 1: The salinity treatment used in the present study may have little impact on the general physiology of seabream. Seabream is a physiological euryhaline species and it can survive a transfer from 30 ppt to 5-6 ppt.  The salinity treatment used in this study was 20 ppt, which is still hyperosmotic to the body fluid of seabream.  In fact, lower salinities at 12-15 ppt are considered as iso-osmotic, where the fish spend less energy on osmoregulation.  Therefore, 20 ppt is not exactly stressful physiologically.  The authors can try a gradual transfer of salinity down to 5-6 ppt (lower than the body fluid osmolality), so that the osmoregulatory epithelia must switch from ion-excretion to ion-absorption.

Response 1: The authors wish to thank the reviewer for the suggestion. In the light of possible acute stress state that fish can experience in aquaculture conditions, it is quite unlikely that salinity can change from 30 ppt to 5 ppt in a fish farm. For this reason, the authors selected a 10 ppt decrease starting from 30 ppt to 20 ppt (which can be considered as a possible unfavourable condition in a fish farm). In this sense the authors wanted to test the effects of a salinity change but within hyperosmotic conditions which is more common in a fish farm.

Re: The reply is not reasonable.  As pointed out previously, 20 ppt cannot be considered as unfavourable condition.  The 30 to 20 ppt transfer did not affect the salt excreting osmoregulatory mechanism of the fish.  If there was any stress, it would be caused by the handling stress.

Point 2: Measurement of IGF-1 and IGF-2 has little physiological meaning as their levels are often not related to the local functions. In addition, the authors should explore the IGF-binding proteins, which modulate the functions (especially local functions) of IGF-1 and IGF-2.  Section 3.2.2 was titled as “Fish growth”.  However, the experiment finished in 60 min, and I do not see any reasonable claims of fish growth under this interval.  I suggest the authors to try a longer acclimation period to induce an observable effect on growth.

Response 2: The section has been changed to fish growth stress markers.

The time course was that short since the authors wanted to test a situation that could happen in a fish farm: if one of the stress situations happens in a fish farm, the personnel will immediately try to fix the problem and thus it is quite unlikely that the stress will last for a long time. In addition, a longer exposure time cannot be considered as “acute stress” which was the target of the present study

Moreover, the most important system for controlling fish growth and development is the somatotropic [growth hormone (GH)-insulin-like growth factor] axis. Studies concerning this axis reported that a functional GH-IGF axis does exist and fits the somatomedin hypothesis, whereby GH promotes increased hepatic IGFs expression. Environmental factors are known to affect the somatotropic axis in fish, and it has been proposed that measurements concerning this axis could provide for an integrated signal, indicative of favourable or unfavourable conditions for fish growth.

As reported in the MS it is well known that a physiological link between stress and growth-related genes exists. Teleost species subjected to various stress factors usually evidence increased serum cortisol levels which have been associated to a suppression of the somatotropic axis. The reviewer proposed an interesting suggestion, however because of the short experimental design (in terms of exposure time-60 min.) the authors decided to analyse the gene expression of both igf 1 and 2 (the reviewer reported IGFs and IGF-binding protein with uppercases which usually refers to the protein and which probably did not significantly change over the short experimental time). On the contrary, gene expression usually undergoes quick variations, easily detectable because of the laboratory technique used (Real time PCR) which may provide information on a possible and further growth delay. As a consequence, the gene expression data presented in the MS should not be considered as a direct fish growth delay but as a negative effect of a short acute stress on the mRNA synthesis of the selected genes and as indicative of favourable or unfavourable conditions for fish growth. Similar results have been reported in other acute stress tests like netting, salinity changes etc performed on other fish species.

Re: The authors are not informative on the roles of IGFBPs on the functions of IGF-1 and IGF-2.  IGFBPs responded before the expression changes of IGFs (e.g. Animal Science Journal 71 (2): 178-188, 2000).  The major drawback on the experimental design is the short treatments (60 min), which cannot be extrapolated to the effects on growth. 

Point 3: The immunohistochemistry in Figure 4 and 5 was chaotic. The legends did not match with the figures.  I suppose A is skin, B is gill, C is stomach, and D is intestine.  For the effects of ammonia and thermal stresses, there are no comparisons with the intact control, nor is there any comparison at all among A-D.  The scale bar should be not correct, as 20 micron is the size of 2 red blood cells, and I think the magnification is way lower to give that resolution.

Response 3: Authors thank Reviewer for the observation. The legends of figures 4 and 5 have been modified to make them clearer to the reader. The main purpose of the figures is to show in which organs/tissues and cell types the HSP70 and IGF-I immunopositivity is located, more than to make comparisons between control and stressed fish. Anyway, a detailed description of the immunoreactivity to both IGF-I and HSP70 and the differences between controls and stressed animals have been added in the results section. Scale bars have been corrected.

Re: The immunohistochemistry results have little meaning if the purpose is to show which organs and cell types possess HSP70 and IGF-1.  Description by text is not a proper presentation for the effects of stresses.  The immunohistochemistry of each treatment for each tissue must be shown to the readers for them to evaluate the degree of changes.  To show the evidence is the basics in a manuscript.

Reviewer 2 Report

MS revisions have been adopted. Be Covid free to continue any work.

Enjoy Christmas time and Have a Happy new Year. 

Reviewer 3 Report

The Authors answered correctly following the reviewer's comments
